# Avoiding catastrophic overfitting in fast adversarial training with adaptive similarity step size

**Jie-Chao Zhao**[1], **Jin Ding**[1,2]*, **Yong-Zhi Sun**[1], **Ping Tan**[1], **Ji-En Ma**[3], **You-Tong Fang**[3]

1 School of Automation and Electrical Engineering & Key Institute of Robotics of Zhejiang Province, Zhejiang University of Science and Technology, Hangzhou, China, 2 State Key Laboratory of Fluid Power and Mechatronic Systems, Zhejiang University, Hangzhou, China, 3 School of Electrical Engineering, Zhejiang University, Hangzhou, China

* jding@zust.edu.cn

**Data Availability Statement:** All relevant data are available at https://doi.org/10.6084/m9.figshare.27991265.v1.

**Funding:** This work was supported by the National Natural Science Foundation of China

## Abstract

Adversarial training has become a primary method for enhancing the robustness of deep learning models. In recent years, fast adversarial training methods have gained widespread attention due to their lower computational cost. However, since fast adversarial training uses single-step adversarial attacks instead of multi-step attacks, the generated adversarial examples lack diversity, making models prone to catastrophic overfitting and loss of robustness. Existing methods to prevent catastrophic overfitting have certain shortcomings, such as poor robustness due to insufficient strength of generated adversarial examples, and low accuracy caused by excessive total perturbation. To address these issues, this paper proposes a fast adversarial training method—fast adversarial training with adaptive similarity step size (ATSS). In this method, random noise is first added to the input clean samples, and the model then calculates the gradient for each input sample. The perturbation step size for each sample is determined based on the similarity between the input noise and the gradient direction. Finally, adversarial examples are generated based on the step size and gradient for adversarial training. We conduct various adversarial attack tests on ResNet18 and VGG19 models using the CIFAR-10, CIFAR-100 and Tiny ImageNet datasets. The experimental results demonstrate that our method effectively avoids catastrophic overfitting. And compared to other fast adversarial training methods, ATSS achieves higher robustness accuracy and clean accuracy, with almost no additional training cost.

## Introduction

Deep learning has become a significant focus of artificial intelligence research in recent years, achieving remarkable results in areas such as autonomous driving [1], intelligent security [2], and smart healthcare [3]. However, these high-performance deep models, particularly deep convolutional neural networks (DCNNs), often exhibit surprising vulnerability when confronted with adversarial attacks. Szegedy *et al.* discovered that minor input perturbations, nearly imperceptible to the human eye, can cause severe classification errors in DCNNs [4].

(No.52293424), and Open Foundation of the State Key Laboratory of Fluid Power and Mechatronic Systems(GZKF-202329). The funders played no role in the study design, data collection and analysis, decision to publish, or preparation of the manuscript.

This phenomenon is not limited to image recognition [5] but also affects other tasks such as speech recognition [6] and natural language processing [7]. The vulnerability of deep learning models to adversarial examples has sparked considerable interest and concern in the field of AI security, leading to the development of various methods aimed at improving model robustness against adversarial attacks.

Adversarial training methods are currently among the most widely used techniques to enhance model robustness [8]. Common adversarial training approaches, such as the Projected Gradient Descent (PGD) adversarial training proposed by Madry *et al.* [9] and the TRadeoff-inspired Adversarial DEfense via Surrogate-loss minimization (TRADES) proposed by Zhang *et al.* [10], employ multi-step iterative processes to generate adversarial examples, which are then used to train the model. While these methods significantly enhance model robustness, they also have notable limitations. In particular, adversarial training for deep neural networks requires multiple forward and backward propagations, leading to substantial additional computational costs.

To address the high computational cost of adversarial training, Wong *et al.* [11] proposed a fast adversarial training method. The main idea is to use the single-step adversarial attack method, Fast Gradient Sign Method (FGSM) [12], instead of the commonly used multi-step attack methods, to generate adversarial examples during training. However, directly using FGSM-generated adversarial examples for training can lead to a phenomenon known as "catastrophic overfitting". Catastrophic overfitting refers to the situation where, during adversarial training with single-step attack methods, the model's performance suddenly becomes very good against single-step adversarial attacks but significantly deteriorates against multi-step attacks [13]. As shown in Fig 1, during traditional FGSM adversarial training, the model's accuracy against multi-step adversarial attacks vanishes as catastrophic overfitting occurs. This issue primarily arises because the model overfits to specific adversarial examples, resulting in high robustness against these particular examples, but nearly completely losing robustness when encountering new adversarial examples.

To prevent catastrophic overfitting, researchers have proposed various improved fast adversarial training methods. Among them, Wong *et al.* introduced the FGSM with Random Start (FGSM-RS) method [11], which adds random noise to clean samples before generating adversarial examples with FGSM. Huang *et al.* introduced Adversarial Training with Adaptive Step Size (ATAS) [14], which adapts the perturbation step size based on the gradient norm during training. Jia *et al.* proposed N-FGSM [15], which adds stronger noise without limiting the total perturbation magnitude to avoid catastrophic overfitting. However, when these methods add random noise and generate adversarial examples using single-step attacks, approaches like FGSM-RS [11] and ATAS [14] often suffer from cancellation between the random noise and adversarial perturbation, resulting in adversarial examples that are not strong enough and leading to insufficient model robustness. On the other hand, N-FGSM [15], due to the excessively large total perturbation, causes difficulty in model convergence and lower accuracy.

In this paper, we propose a fast adversarial training method based on adaptive similarity step size. After adding random noise to the original image samples for data augmentation, the samples are input into the model to compute their gradient information. The perturbation step size of the adversarial examples is then adaptively adjusted based on the similarity between the noise and the gradient direction. We design a similarity algorithm that combines Euclidean distance similarity and cosine similarity to evaluate the similarity between the noise and the gradient direction, thereby generating adversarial examples that are more conducive to effective adversarial training. The experimental results demonstrate that our method successfully avoids catastrophic overfitting while achieving high robustness accuracy and clean accuracy.

The major contributions of this paper are two-folds:

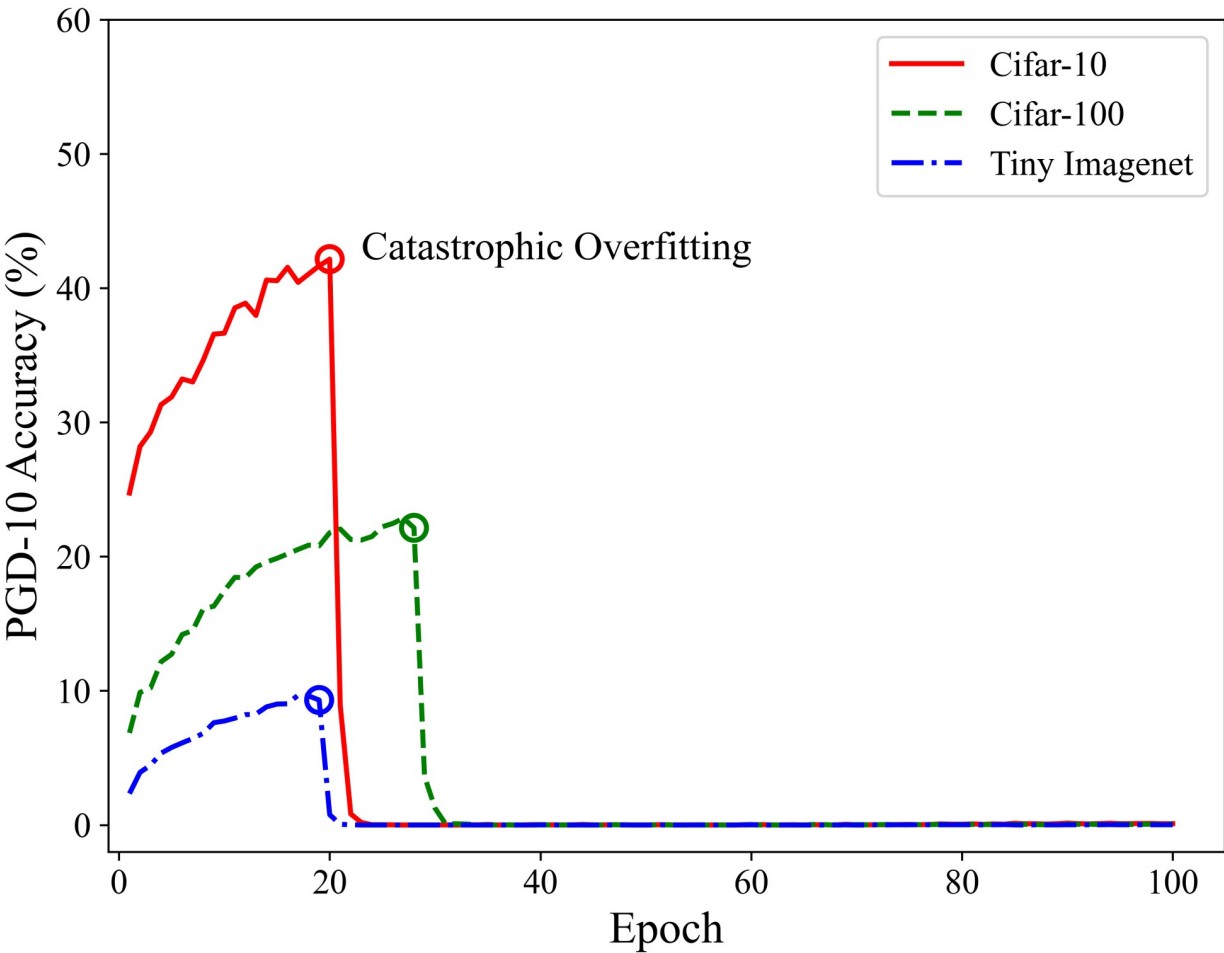

**Fig 1. Adversarial training process where catastrophic overfitting occurs.**

- We propose an effective fast adversarial training method—fast adversarial training with adaptive similarity step size (ATSS for short). The core of this method lies in the design of an adaptive step size algorithm, which first calculates the similarity between random noise and the gradient direction using Euclidean distance and cosine similarity. Then, the required perturbation step size is computed based on this similarity, making the step size inversely proportional to the similarity. Finally, adversarial examples are generated based on the calculated step size and used for adversarial training, thereby enhancing the robustness of the target model.

- We conducted multiple adversarial attack tests on the ResNet18 and VGG19 models using the CIFAR-10, CIFAR-100 and Tiny ImageNet datasets, employing attack methods including FGSM, PGD, C&W, and AA. The experimental results demonstrate that ATSS effectively avoids catastrophic overfitting. And compared to other fast adversarial training methods, ATSS achieves higher robustness accuracy and clean accuracy, with almost no additional training cost.

## Related work

### Adversarial examples and adversarial attack methods

Adversarial example is a key concept in the field of deep learning robustness, first introduced by Szegedy *et al.* in [4]. Adversarial examples are specially crafted samples designed to deceive deep learning models by introducing subtle, carefully calculated perturbations to the original inputs. These perturbations can cause the model to produce incorrect classification results, despite being nearly imperceptible to the human eye.

Gradient-based adversarial attack methods are the most commonly used techniques for generating adversarial examples. Among them, Goodfellow *et al.* [12] introduced FGSM. This method begins by calculating the gradient of the model's loss function with respect to the input sample *x*. The sign of this gradient is then multiplied by a perturbation budget $\varepsilon$. The resulting value is added to the original image to generate the adversarial example $\boldsymbol{x}'$, represented as:

$$\boldsymbol{x}' = \boldsymbol{x} + \epsilon \cdot \text{sign}(\nabla_x L(\boldsymbol{x}, y)) \tag{1}$$

where $\nabla_x$ is the gradient with respect to *x*. Madry *et al.* proposed PGD [9], which is an iterative adversarial attack technique. Due to its strong attack capability and broad applicability, PGD is regarded as the benchmark method for generating adversarial examples. PGD can be understood as an iterative version of FGSM, where a smaller perturbation step size is used in each iteration. Additionally, a projection operation is included at each step to ensure that the adversarial examples remain within the predefined perturbation budget $[-\varepsilon, \varepsilon]$, represented as:

$$\boldsymbol{x}'_{i+1} = \text{clip}_{\boldsymbol{x},\varepsilon}\{\boldsymbol{x}'_i + \alpha \cdot \text{sign}(\nabla_{\boldsymbol{x}'_i} L(\boldsymbol{x}'_i, y))\} \tag{2}$$

where *i* is the step number in PGD, and clip($\cdot$) is the clipping function that ensures the generated adversarial examples satisfy both the perturbation budget $\varepsilon$ constraint and the image's pixel value range. Building on PGD, researchers have developed various iterative attack methods [16–18]. Additionally, there are optimization-based adversarial attack methods, such as Carlini & Wagner attack (C&W) proposed by Carlini *et al.* [19]. This method generates adversarial examples by solving a constrained optimization problem, aiming to find the minimal perturbation that leads to the misclassification of the sample.

Croce *et al.* [20] introduced the AutoAttack (AA) method, a comprehensive adversarial attack testing framework. AutoAttack combines multiple advanced attack methods, including Adaptive PGD with Cross-Entropy loss (APGD-CE) [20], Adaptive PGD with Decision-Based loss Reversion (APGD-DLR) [20], Fast Adaptive Boundary attack (FAB) [21], and Square Attack [22], with the goal of systematically evaluating the robustness of neural network models.

### Adversarial defense methods

To counter the threat posed by adversarial examples, researchers have proposed various defense methods against adversarial attacks. Common approaches include adversarial training [10], input detection [23], preprocessing [24], defensive distillation [25], and gradient masking [26].

Among these methods, adversarial training (AT) has garnered the most attention. The fundamental idea of AT is to train the model using adversarial examples generated through adversarial attacks during the training process. This allows the model to gradually adapt to these complex and adversarial examples, thereby improving its robustness. Mathematically, this approach involves solving a min-max optimization problem, which can be expressed as

follows:

$$\min_{\theta} \mathbb{E}_{(x,y) \sim D} \left[ \max_{\Delta x \in \varepsilon} L(f_\theta(x + \Delta x), y) \right] \qquad (3)$$

where $D$ represents the training dataset, $x$ is the input sample, $y$ is the label, $\Delta x$ is the adversarial perturbation, $\varepsilon$ is the perturbation budget, $L(\cdot)$ is the loss function, and $f_\theta$ represents the deep learning model with parameters $\theta$. The ultimate goal is to optimize the model parameters $\theta$ to enhance the adversarial robustness of the DCNNs.

Madry *et al.* proposed the PGD-AT method [9], which is one of the most successful adversarial training methods. This approach uses the PGD method to iteratively generate adversarial examples for training, thereby improving the robustness of deep learning models. Building on this, Zhang *et al.* introduced the TRADES adversarial training method [10], which redesigned the loss function with a particular emphasis on enhancing the model's adversarial robustness while maintaining classification accuracy.

Due to the high computational cost required by multi-step adversarial training methods, Shafahi *et al.* proposed the Free Adversarial Training (Free-AT) [27] method. This approach involves multiple iterations of training on the same mini-batch of samples, where the gradient information obtained from the previous iteration is reused to update both the model parameters and the adversarial examples. This eliminates the additional computational cost of generating adversarial examples. However, since Free-AT requires multiple iterations on each mini-batch, the training time remains long, and the resulting model's robustness is still insufficient. Consequently, researchers began exploring the use of single-step adversarial attack methods in adversarial training to replace multi-step methods for generating adversarial examples, leading to the development of fast adversarial training methods [11].

However, simply using single-step adversarial attack methods for adversarial training can result in catastrophic overfitting, where the model loses robustness. To address this issue, Wong *et al.* proposed the FGSM-RS [11] method, which involves adding random noise to the input sample before generating adversarial perturbations, represented as:

$$x_{noise} = x + \text{Uniform}(-k\varepsilon, k\varepsilon) \qquad (4)$$

where $k$ is the noise coefficient, representing the noise range of $[-k\varepsilon, k\varepsilon]$, and $x_{noise}$ is the sample after adding random noise.

Similarly, Jia *et al.* proposed the Fast Gradient Sign Method with prior from the Momentum of all Previous Epoch (FGSM-MEP) [28], which introduces a momentum mechanism that accumulates the adversarial perturbations generated in previous iterations and uses them as the initial perturbation for the next iteration, with periodic resets after a certain number of iterations. Although these methods can partially mitigate catastrophic overfitting, they still lag behind multi-step adversarial training methods in terms of robustness and can still experience catastrophic overfitting when the perturbation budget is large. And Riushchenko *et al.* introduced a regularizer called GradAlign [29], which aligns the gradient directions of the model for clean samples and their corresponding adversarial examples during adversarial training. This alignment helps prevent the neural network model from exhibiting locally highly nonlinear behavior, thereby avoiding catastrophic overfitting. However, since GradAlign requires multiple gradient computations, its training efficiency is lower than that of FGSM-RS. Huang *et al.* proposed a fast adversarial training method with adaptive step size adjustment, known as ATAS [14]. The core of this method lies in adjusting the step size based on the $l_2$ norm of the gradient for the training samples. Jorge *et al.* proposed N-FGSM [15], which effectively avoids catastrophic overfitting and further enhances model robustness by adding stronger noise to

the original clean samples and not restricting the total perturbation during training. However, the drawback of this approach is that the excessive perturbation leads to lower model accuracy.

Existing fast adversarial training methods generally suffer from issues such as weak robustness, low accuracy, and a tendency toward catastrophic overfitting. To address these problems, we improve upon existing methods and propose the ATSS method. We design a similarity evaluation approach and calculate the required perturbation step size based on the similarity between the random noise and the gradient direction. Experimental results show that this method can effectively prevent catastrophic overfitting and achieve high accuracy on both adversarial examples and clean samples.

## Fast adversarial training with adaptive similarity step size

Currently, fast adversarial training methods that generate adversarial examples using the single-step adversarial method FGSM have garnered widespread attention. Compared to traditional multi-step adversarial training methods, this approach significantly reduces the required computational cost. However, it also has some drawbacks, with catastrophic overfitting being the most prominent issue, leading to a loss of model robustness. As shown in Fig 2, when using the traditional single-step attack method FGSM to perform adversarial training on a ResNet18 model, catastrophic overfitting occurred after 20 epochs of training, resulting in an

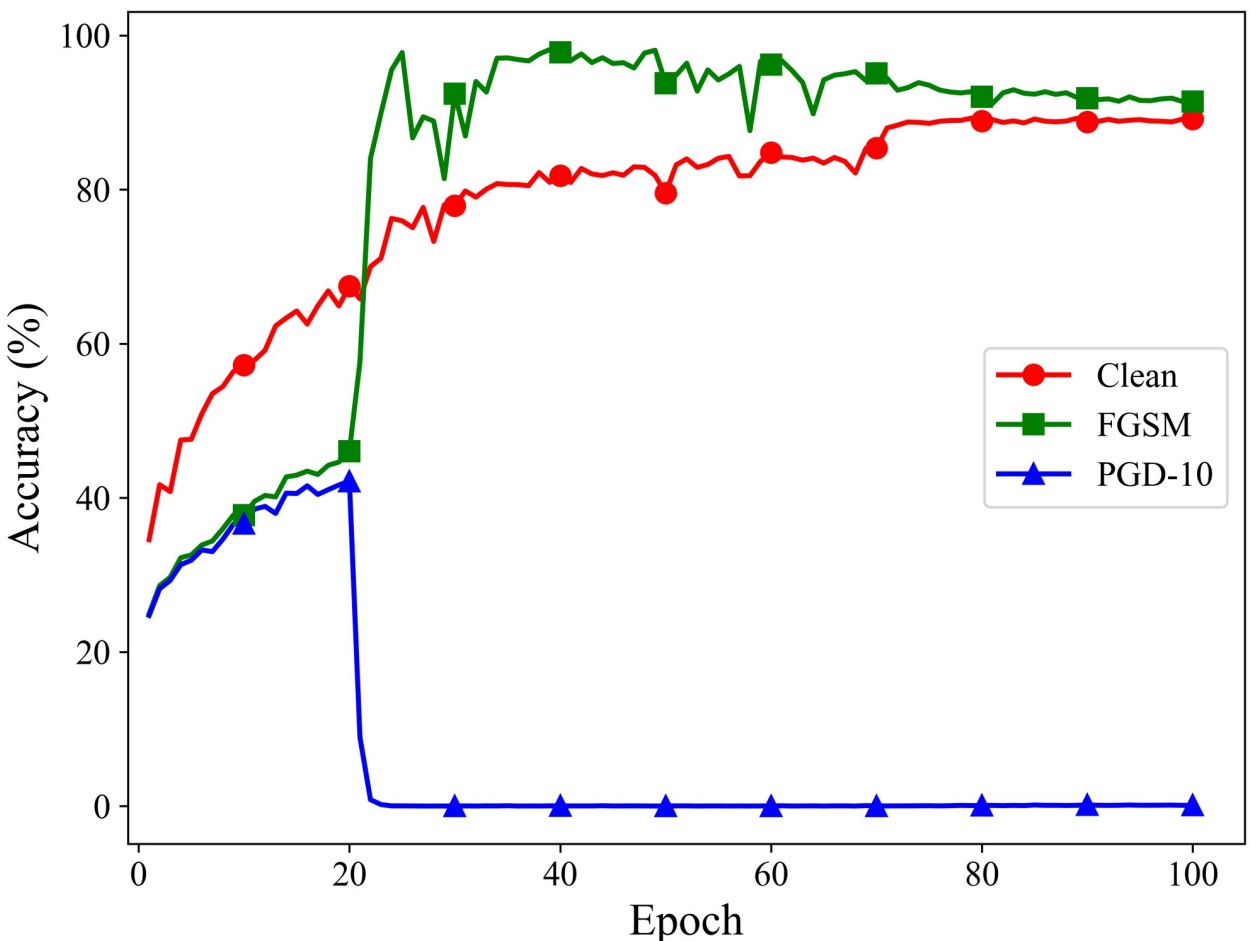

**Fig 2. Adversarial training processes in traditional FGSM adversarial training.**

**Table 1. Attack success rates of different single-step attack methods.**

| Attack methods | Attack success rate |
|---|---|
| FGSM | 48.48 |
| FGSM-RS($\alpha = 8/255$) | 42.36 |
| FGSM-RS($\alpha = 10/255$) | 45.10 |

almost complete loss of defense against multi-step attacks. Although several methods have been developed to avoid this phenomenon, they generally suffer from either insufficient robustness or inadequate classification accuracy on clean samples.

To address these issues, this paper proposes a fast adversarial training method based on adaptive similarity step size, which aims to overcome the problem of catastrophic overfitting while simultaneously improving robust accuracy and clean accuracy.

## Motivation

One of the main reasons for the occurrence of catastrophic overfitting is that the magnitude of the perturbations in the generated adversarial examples is overly uniform [30]. To enhance the diversity of the training data and avoid catastrophic overfitting, existing fast adversarial training methods generally use random noise initialization. This involves adding random noise perturbations to the input samples before generating adversarial perturbations. The samples with added random noise are then input into the model to compute gradients and obtain adversarial perturbations. The sum of the random perturbations and the adversarial perturbations constitutes the total perturbation.

Adding random noise for initialization can change the magnitude of the total perturbation, thereby avoiding catastrophic overfitting. In FGSM-RS, the strategy is adopted where a clipping function is used to limit the total perturbation within a preset perturbation budget. This method can prevent the total perturbation from becoming excessively large.

We find that, attack methods using random noise initialization and limiting the total perturbation size exhibit a significant decrease in attack strength compared to the original FGSM attack method. Table 1 compares the attack success rates of the attack methods used by FGSM-RS and the original FGSM attack method against a ResNet18 model trained with TRADES adversarial training, with the maximum perturbation limit set to 8/255. In FGSM-RS, the step size is set to 8/255 and 10/255, respectively. From the table, it can be seen that the attack success rate of the FGSM-RS method is significantly lower than that of the original FGSM attack method in both settings of step size. This also leads to the reduced robustness of fast adversarial training methods like FGSM-RS [11] and FGSM-MEP [28], which generate adversarial examples using attack methods that employ random noise initialization and limit the total perturbation size.

The main reason for the reduced attack strength of FGSM after random initialization is that the direction of the random perturbation partially opposes the gradient direction, causing the random perturbation and the adversarial perturbation to cancel each other out. If the maximum perturbation is further limited, this results in a decreased average total perturbation. Table 2 compares the average total perturbation magnitudes of adversarial examples generated by the attack methods used in FGSM-RS and the original FGSM attack method, with the maximum perturbation limit set to 8/255. In FGSM-RS, the step size is set to 8/255 and 10/255, respectively. From the table, it can be seen that the average total perturbation magnitude of adversarial examples generated by FGSM-RS in both settings of step size is less than the perturbation limit, which leads to insufficient strength of the generated adversarial examples.

**Table 2. Average total perturbation magnitudes generated by different single-step attack methods.**

| Attack methods | Average perturbation |
|---|---|
| FGSM | 8.000/255 |
| FGSM-RS($\alpha = 8/255$) | 6.000/255 |
| FGSM-RS($\alpha = 10/255$) | 6.875/255 |

**Table 3. Maximum total perturbation magnitudes generated by different single-step attack methods.**

| Attack methods | Maximum perturbation |
|---|---|
| FGSM | 8.000/255 |
| FGSM-RS | 8.000/255 |
| N-FGSM | 24.000/255 |

In N-FGSM [15], larger noise is used to initialize the samples when generating adversarial examples, and the maximum perturbation is not limited, ensuring the attack strength of the adversarial examples. However, the total perturbation magnitude of adversarial examples generated by N-FGSM is excessively large, which leads to lower classification accuracy of the model. Table 3 compares the maximum total perturbation magnitudes of the attack methods used in N-FGSM with those generated by FGSM-RS and FGSM, all under a perturbation budget of 8/255. From the table, it is evident that the maximum total perturbation magnitude of the adversarial examples generated by N-FGSM significantly exceeds the perturbation budget, resulting in insufficient clean accuracy for the model.

Based on the above analysis, we propose a new fast adversarial training method called ATSS. Unlike traditional methods that use simple clipping to limit the perturbation magnitude, ATSS adaptively adjusts the perturbation step size based on the similarity between the random noise and the gradient direction, making the step size inversely proportional to the similarity. When the similarity is low, the perturbation step size is increased to ensure the attack strength of the adversarial examples; when the similarity is high, the step size is decreased to prevent the total perturbation from becoming too large. A comparison of adversarial examples generated by different methods is shown in Fig 3.

**ATSS.** To ensure that the adversarial examples generated during fast adversarial training maintain both attack strength and diversity while limiting excessive perturbation, this paper proposes a fast adversarial training method based on adaptive similarity step size. The flowchart of this method is illustrated in Fig 4.

After extracting an initial batch of samples $x$ from the dataset, a random noise $\eta$ with the same shape as $x$ is generated, where the values of $\eta$ are within the range [-1,1]. This random noise is then scaled and added to the initial samples. The augmented samples are subsequently input into the model to compute the gradient direction $v$, represented as:

$$\eta = \text{Uniform}(-1, 1) \tag{5}$$

$$x_{\text{noise}} = x + \varepsilon \cdot \eta \tag{6}$$

$$v = \text{sign}(\nabla_x L(f(x_{\text{noise}}), y)) \tag{7}$$

where $\varepsilon$ is the perturbation budget, $f$ is the target model.

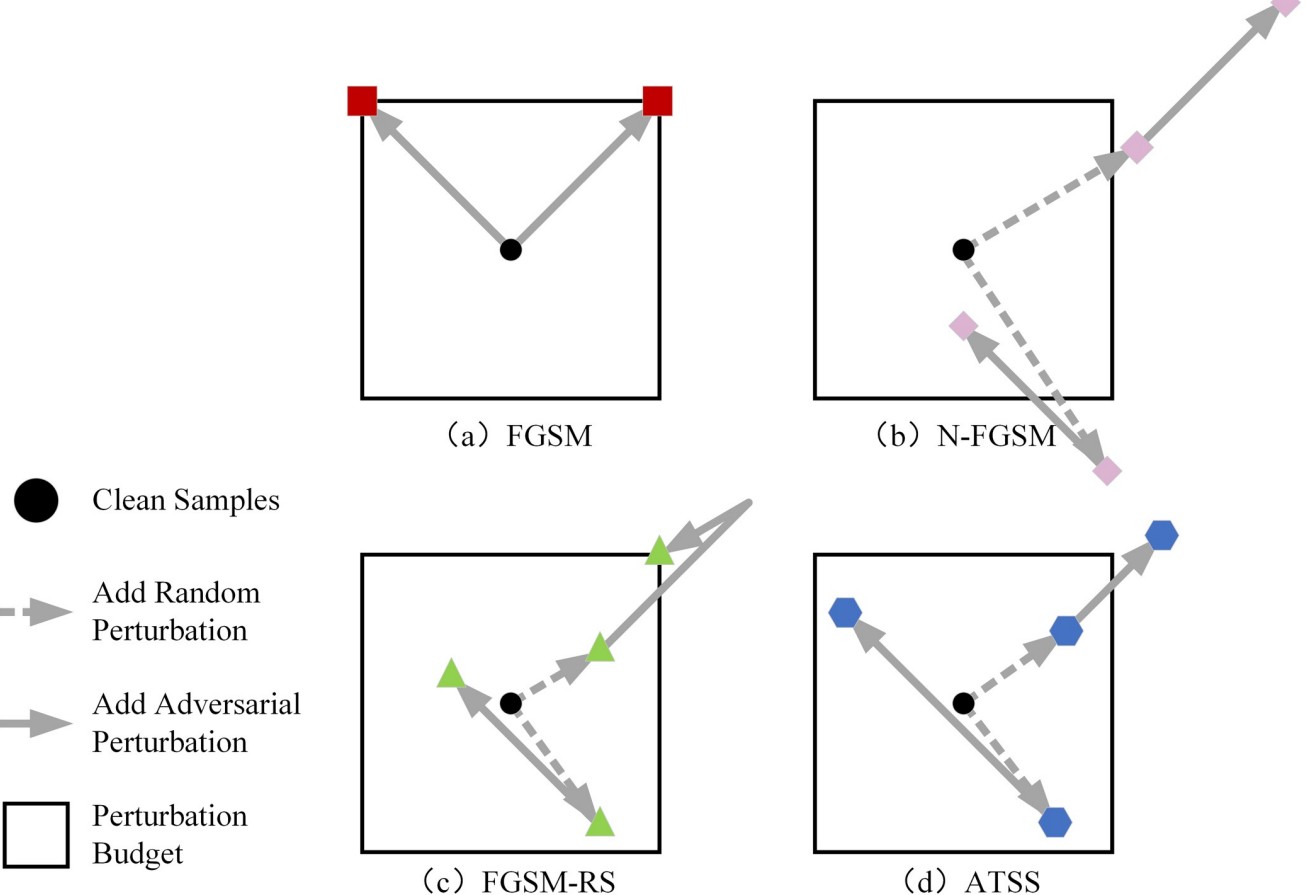

**Fig 3. Comparison of adversarial samples generated by different single-step attack methods.**

Next, the similarity between the added noise $\eta$ and the gradient direction $v$ is calculated. To evaluate the similarity between two vectors, both their magnitude and directional similarities should be assessed simultaneously. As shown in Fig 5(a), using only Euclidean distance similarity can neglect differences in direction. Similarly, in Fig 5(b), using only cosine similarity will neglect differences in magnitude (In (a), the Euclidean distance similarity between $v$ and $\eta_1$ is the same as that between $v$ and $\eta_2$. In (b), the cosine similarity between $v$ and $\eta_1$ is the same as that between $v$ and $\eta_2$.). To better measure the similarity between the two vectors, taking into account both magnitude and direction information, this paper uses a method that combines Euclidean distance similarity and cosine similarity to evaluate the overall similarity.

Firstly, calculate the Euclidean distance $D$ and cosine value $C$ between $\eta$ and $v$. represented as:

$$D(\eta, v) = \sqrt{\sum_{i=1}^{n}(\eta_i - v_i)^2} \tag{8}$$

$$C(\eta, v) = \cos(\eta, v) = \frac{\sum_{i=1}^{n} \eta_i \cdot v_i}{\|\eta\| \cdot \|v\|} \tag{9}$$

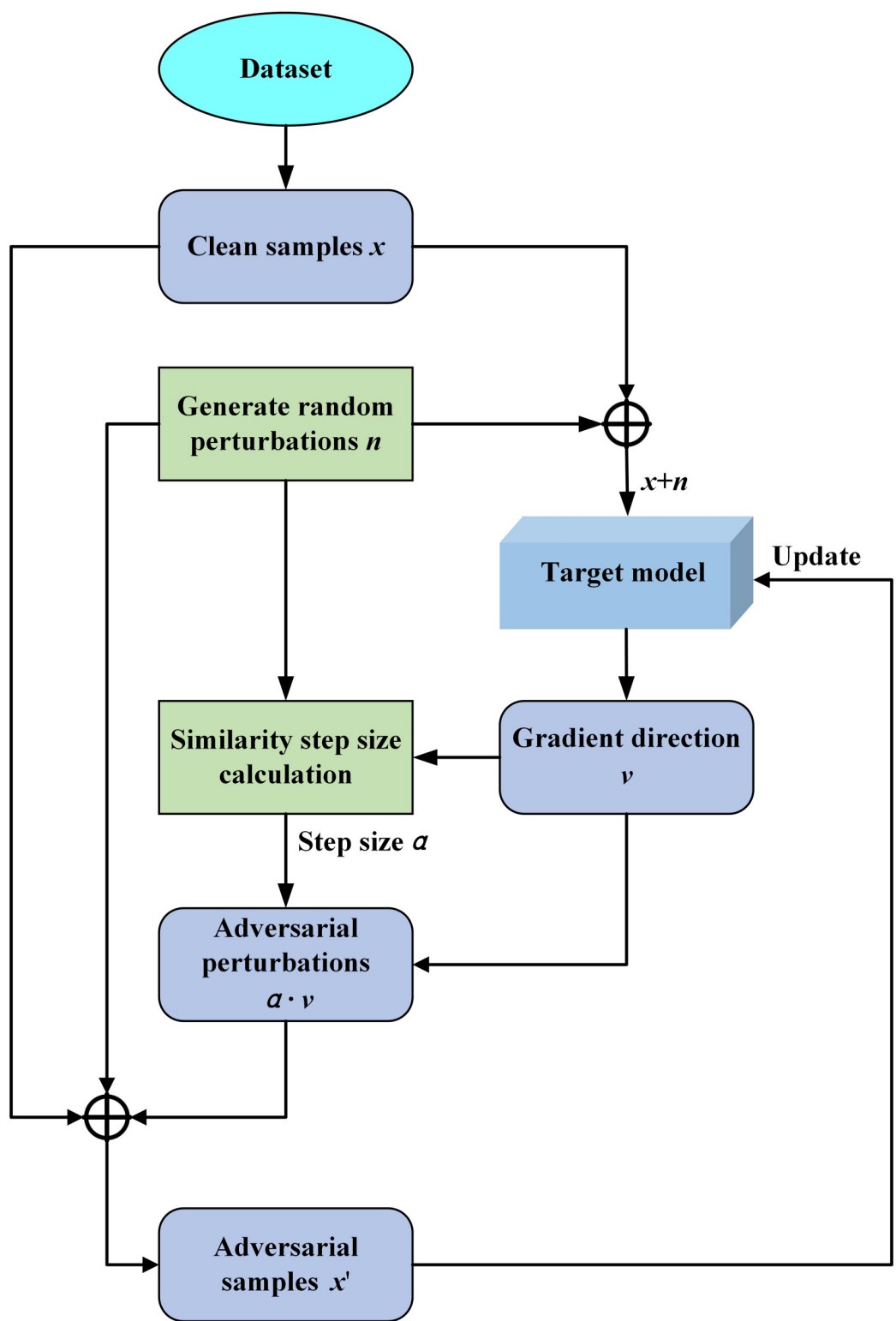

**Fig 4. Flowchart of fast adversarial training method based on adaptive similarity step size.**

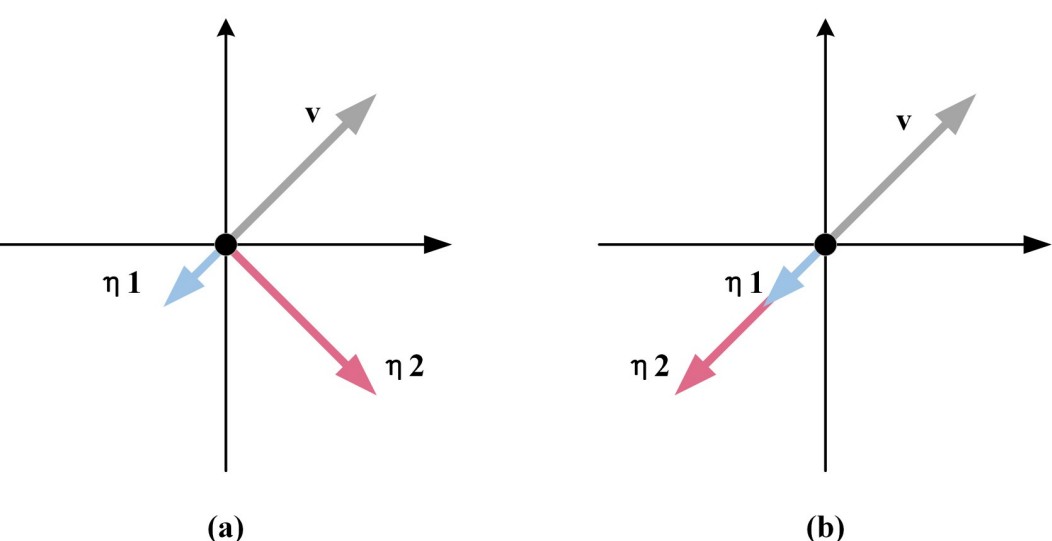

**Fig 5. Illustrative examples for Euclidean distance similarity and cosine similarity.**

where $n$ is the dimension of the vectors, and $\eta_i$ and $v_i$ are the $i$-th elements of $\eta$ and $v$, respectively.

Then, normalize them to obtain the Euclidean distance similarity $S_{ed}$ and cosine similarity $S_{cos}$, represented as:

$$S_{ed}(\eta, v) = -\frac{D(\eta, v) - mean(D)}{std(D)} \tag{10}$$

$$S_{cos}(\eta, v) = \frac{C(\eta, v) - mean(C)}{std(C)} \tag{11}$$

where $mean(D)$ is the mean of the Euclidean distances, $std(D)$ is the standard deviation of the Euclidean distances, $mean(C)$ is the mean of the cosine values, and $std(C)$ is the standard deviation of the cosine values.

The similarity $s$ is obtained by adding the Euclidean distance similarity and cosine similarity. Then, the perturbation step size $\alpha$, which is inversely proportional to $s$ is generated, represented as:

$$s = S_{ed}(\eta, v) + S_{cos}(\eta, v) \tag{12}$$

$$\alpha = (1 - \beta \cdot s) \cdot \alpha_0 \tag{13}$$

where $\beta$ is the influence coefficient and $\alpha_0$ is the standard step size.

The adversarial perturbations are generated based on the step size $\alpha$, and they are added to $x_{noise}$ to obtain the adversarial examples. The calculation formula for the adversarial example is as follows:

$$x' = x_{noise} + \alpha \cdot v \tag{14}$$

Finally, the generated adversarial example $x'$ is fed into the model to update the model parameters. The pseudocode for the ATSS algorithm is presented in Algorithm1.

**Algorithm 1** ATSS

```
Input: Number of training epochs N, perturbation budget ε, dataset D,
samples x, labels y, size of each training batch M, influence coeffi-
cient β, standard step size α₀, loss function L, target model f and its
parameters θ, learning rate γ
Output: Optimized model f_θ
1:  for t = 1 to N do
2:    for i = 1 to M do
3:       η_i ← Uniform(-1, 1)
4:       x_noise,i ← x_i + ε · η_i;
5:       v_i ← sign(∇_{x_i} L(f(x_noise,i), Y_i))
6:       s ← S_ed(η_i, v_i) + S_cos(η_i, v_i)
7:       α ← (1 - β · s) · α₀
8:       x'_i ← x_noise,i + α · v_i
9:       θ ← θ - γ∇_θ L(f(x'_i), Y_i)
10:     end for
11:  end for
```

## Experiments

To validate the practicality of the proposed method, we conduct experiments to assess whether ATSS can achieve a model with both strong robustness and high classification accuracy while maintaining a relatively low computational cost. We compare ATSS with several adversarial training methods. All experiments are performed on a platform equipped with a 3.0GHz i9-13900k CPU, 128GB of RAM, and an RTX 4080 GPU, using the PyTorch framework.

### Datasets

In the experimental section, we use three publicly available benchmark datasets: CIFAR-10 [31], CIFAR-100 [31], and Tiny ImageNet [32]. CIFAR-10 and CIFAR-100 were released by the Canadian Institute for Advanced Research (CIFAR). The CIFAR-10 dataset contains a total of 60,000 images across 10 categories, such as "airplan" and "car", with 50,000 images used for training and 10,000 for testing. The CIFAR-100 dataset is similar to CIFAR-10 but includes 60,000 images divided into 100 categories, with the same 50,000 images used for training and 10,000 for testing. Both datasets consist of RGB images with a size of 32×32×3. The Tiny ImageNet dataset is a smaller version of the ImageNet dataset [33], divided into 200 categories, containing 100,000 training images and 10,000 testing images. The Tiny ImageNet dataset consists of RGB images with a size of 64×64×3.

### Model parameters

For the experimental setup, we select ResNet18 [34] and VGG19 [35] as the target models for testing, as these models are widely used in various image recognition tasks. The models do not use any pre-trained weights. The optimizer used in the experiments is the SGD optimizer, with a momentum of 0.9 and a weight decay coefficient of 5e-4. The initial learning rate is set to 0.01, and the batch size for training samples is set to 128. The influence coefficient $\beta$ is set to 0.04, the standard step size $\alpha_0$ is set to 10/255 (see Section Hyperparameter Experiments).

### Attack methods

To verify the generalizability of the proposed method, we employ several commonly used adversarial attack methods to attack the resulting models, including FGSM [12], PGD [9], C&W [19], and AA [20]. In the experiments, the perturbation budget $\varepsilon$ for all adversarial attacks is set to 8/255. The step size $\alpha$ for each PGD attack is set to 2/255, with the number

following PGD indicating the number of iterations (e.g., PGD-10 represents 10 steps of PGD). The C&W adversarial attack method used is based on modifying the C&W loss with PGD, with 20 iterations.

## Baselines

To verify the effectiveness of the proposed method, we compare it with other fast adversarial training methods: FGSM-RS [11], N-FGSM [15], FGSM-MEP [28], ATAS [14], and FGSM-GA [29]. The experimental parameters for the other fast adversarial training methods are set according to the configurations provided in their respective papers and source codes.

## Metrics

The metrics we use in the experiments is the classification accuracy. The formula for calculating classification accuracy is as follows:

$$r = \frac{p}{t} \cdot 100\% \tag{15}$$

where $r$ represents the classification accuracy, $t$ represents the total number of test examples, and $p$ represents the total number of examples correctly predicted by the model.

## Experiments on preventing catastrophic overfitting

To verify that the proposed method can effectively prevent catastrophic overfitting, we conduct long-term adversarial training over 100 epochs on the CIFAR-10 dataset using ResNet18 and VGG19 as the target models. After each training epoch, we test the models' classification accuracy on clean samples, under FGSM attacks, and under PGD-10 attacks. The experimental results are shown in Figs 6 and 7.

The results indicate that the proposed fast adversarial training method based on adaptive similarity step size effectively prevents catastrophic overfitting. Throughout the 100 epochs of adversarial training, both the ResNet18 and VGG19 models maintain good classification accuracy when facing the single-step attack method FGSM and the multi-step attack method PGD.

We also perform the same tests on the CIFAR-100 and Tiny ImageNet datasets. The model used is ResNet18. On both datasets, ATSS can effectively avoid catastrophic overfitting, shown in Figs 8 and 9.

Additionally, we compare the clean accuracy and robust accuracy during the training process of ATSS, FGSM-RS [11], and N-FGSM [15] on ResNet18 model and CIFAR-10 dataset. As shown in Fig 10, compared to the other two fast adversarial training methods, ATSS demonstrates better performance during the training process.

## Experiments on adversarial robustness and training cost

We evaluate the adversarial robustness and the training time required for the models trained using the proposed method. To validate the effectiveness of our method, we compare it with five other fast adversarial training methods: FGSM-RS [11], N-FGSM [15], FGSM-MEP [28], ATAS [14], and FGSM-GA [29]. For reference, we also test two traditional multi-step adversarial training methods, PGD-10-AT [9] and TRADES [10]. The experimental parameters for the other adversarial training methods are set according to the configurations provided in their respective papers and source codes. In FGSM-RS, the step size is set to 10/255.

All methods are trained for 60 epochs, and the total training time is recorded. For robustness testing, we use five attack methods, i.e., FGSM, PGD-10, PGD-50, C&W, and AA, to

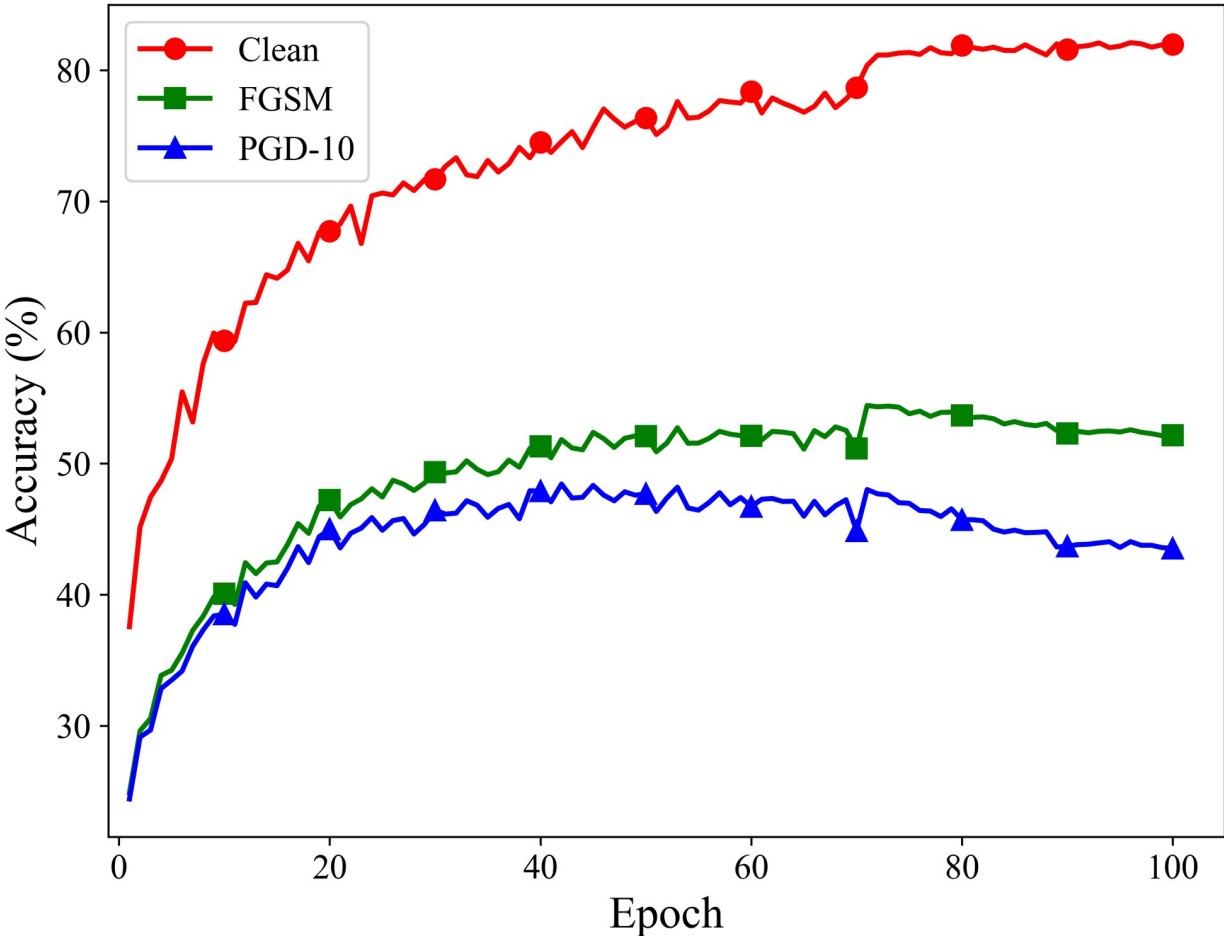

**Fig 6. Adversarial training process of ATSS on the ResNet18 model.**

attack the trained models and record the classification accuracy under these attacks. The detailed experimental results are shown in Table 4.

As shown in Table 4, compared to other fast adversarial training methods, our method consistently achieves higher classification accuracy under all multi-step attack methods. Additionally, the classification accuracy on clean samples remains at a high level. In terms of training time, since our method does not require additional backpropagation compared to methods like FGSM-RS, the training duration is similar to other fast adversarial training methods.

When comparing ATSS with the traditional multi-step adversarial training method TRADES, the proposed method demonstrates comparable robust accuracy and much faster training process, saving more than 80% training time, shown in Fig 11.

To further validate the effectiveness of our method across different datasets, we conduct additional tests on the CIFAR-100 and Tiny ImageNet datasets using the PGD-10 attack method to evaluate classification accuracy. The target models are compared against the five fast adversarial training methods. The experimental results are shown in Table 5. As can be seen from Table 5, our method also maintains better robustness on both the CIFAR-10 and Tiny ImageNet datasets compared to other fast adversarial training methods.

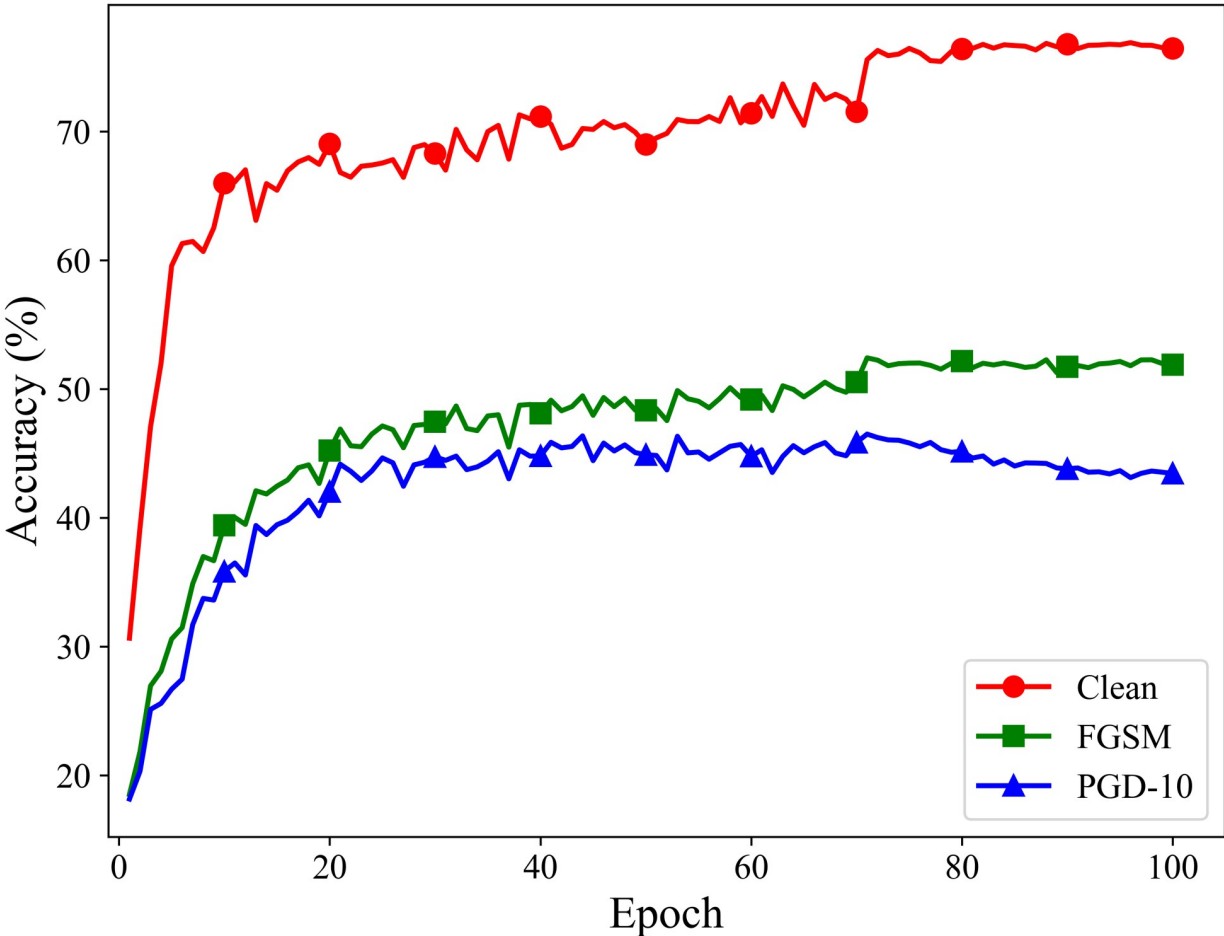

**Fig 7. Adversarial training process of ATSS on the VGG19 model.**

### Ablation study

To better assess the similarity between two vectors while considering both magnitude and direction information, we combine Euclidean distance similarity and cosine similarity to evaluate overall similarity and generate the perturbation step size. To demonstrate the effectiveness of this method, we compare it with strategies that use only Euclidean distance similarity or only Euclidean distance for generating the step size, conducting an ablation study on CIFAR-10 and ResNet18. The experimental results, shown in Table 6, indicate that combining Euclidean distance similarity and cosine similarity for generating perturbation step sizes leads to better robustness in adversarial training.

### Hyperparameter experiments

To illustrate the impact of the various hyperparameters in the proposed method and to identify the most suitable parameter values, we conduct the experiments on CIFAR-10, using ResNet18.

**Influence coefficient.** As shown in Eq (13), the influence coefficient $\beta$ is used to control to which extent similarity affects the perturbation step size. If $\beta$ is set to 0, the generated perturbation step size equals the standard step size.

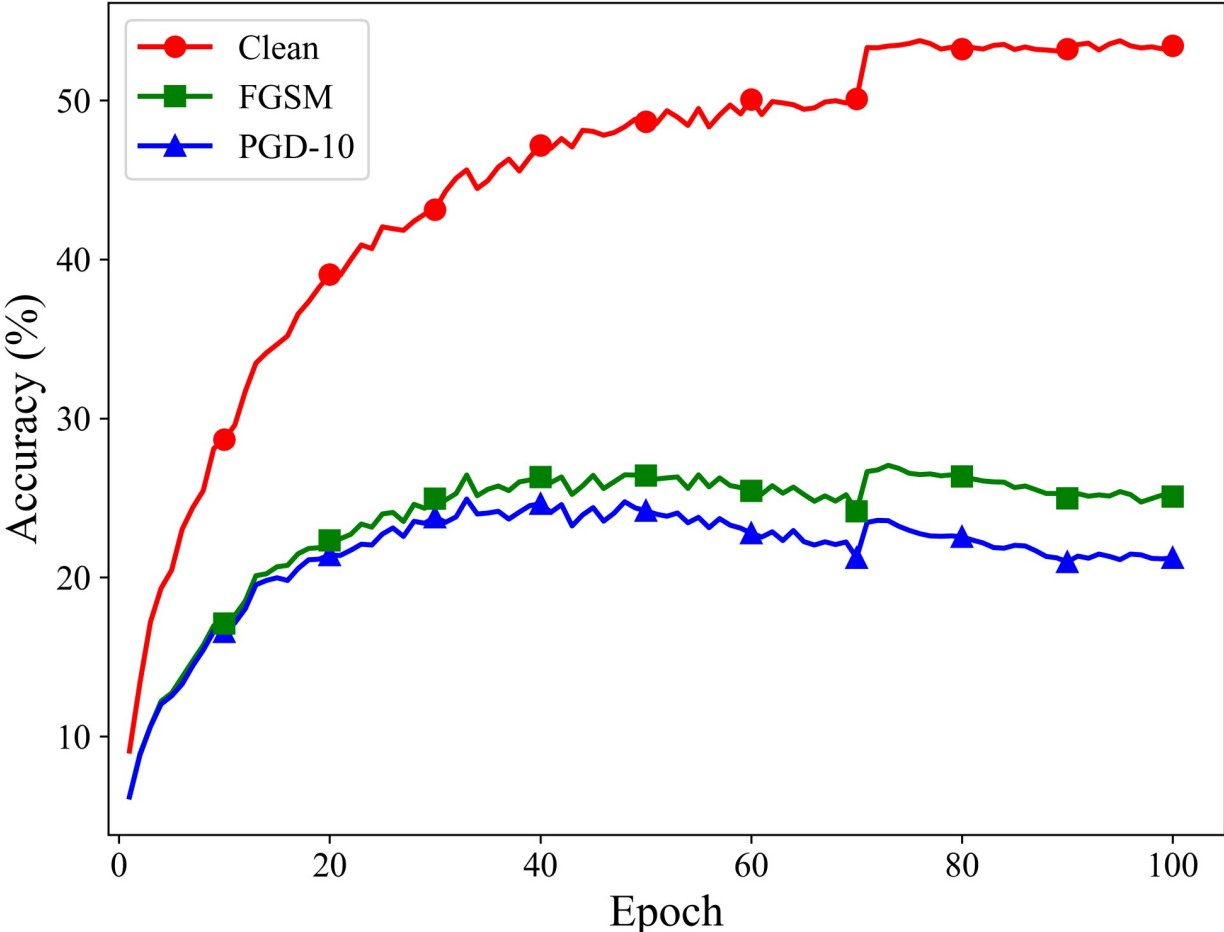

**Fig 8. Adversarial training process of ATSS on the CIFAR-100 dataset.**

Here, we test the classification accuracy of models trained with different influence coefficients under adversarial attacks. The experimental results are shown in Table 7.

The results indicate that setting the influence coefficients too high or too low can also lead to decreased robustness. Therefore, in our experiments, the influence coefficients $\beta$ is set to 0.04.

**Standard step size.** As shown in Eq (13), the standard step size $\alpha_0$ controls the magnitude of the average generated adversarial perturbation. Here, we test the classification accuracy of models trained with different standard step size under adversarial attacks. The experimental results are shown in Table 8.

The results indicate that as the value of the standard step size increases, the robustness of the model gradually improves. However, excessively large standard step size can lead to instability during model training. Therefore, in our experiments, the standard step size $\alpha_0$ is set to 10/255.

## Discussions

The proposed fast adversarial training method based on ATSS can effectively enhance the ability of deep convolutional neural networks to resist adversarial image examples. It is

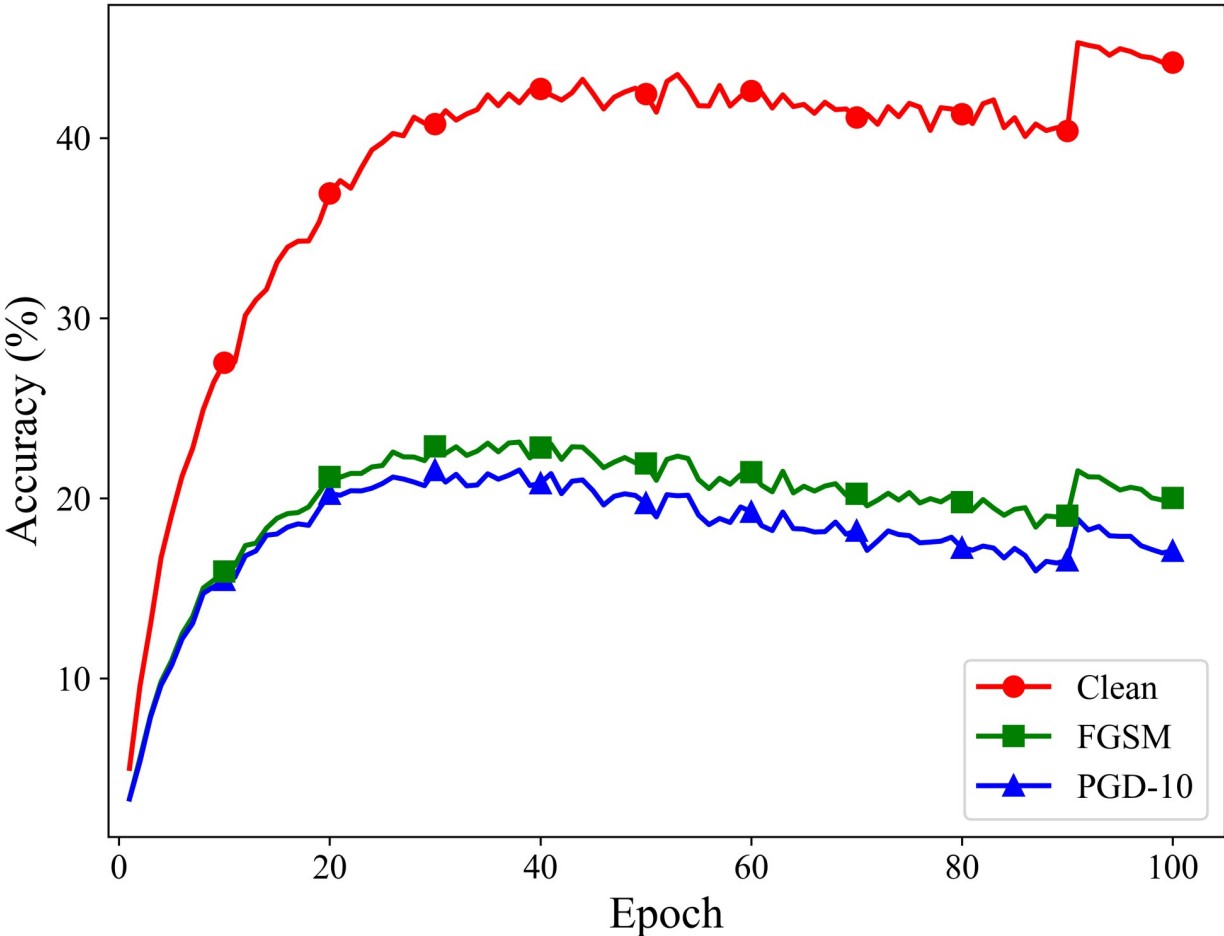

**Fig 9. Adversarial training process of ATSS on the Tiny ImageNet dataset.**

noteworthy that, this technique can be readily extended to adversarial defense for speech [6] and video [36] data. In speech and video classification, recurrent neural networks are a commonly used method [37]. Similar to convolutional neural networks, adversarial training techniques can also be applied to recurrent neural networks to enhance their robustness against adversarial examples [38]. ATSS, as an acceleration technique for adversarial training and avoiding catastrophic overfitting, can be seamlessly applied to recurrent neural networks to enhance their robustness to adversarial speech and video examples.

## Conclusion

To improve the efficiency of adversarial training and enhance the robustness of models while avoiding catastrophic overfitting during training, we propose a fast adversarial training method with adaptive similarity step size (ATSS for short). This method involves first initializing the samples with random noise and then feeding them into the model to obtain gradient information. The perturbation step size for each sample is calculated based on the similarity between the added random noise and the gradient direction. This approach ensures the diversity and attack strength of the adversarial examples while preventing excessive perturbations, thereby generating adversarial examples that are more conducive to effective model training.

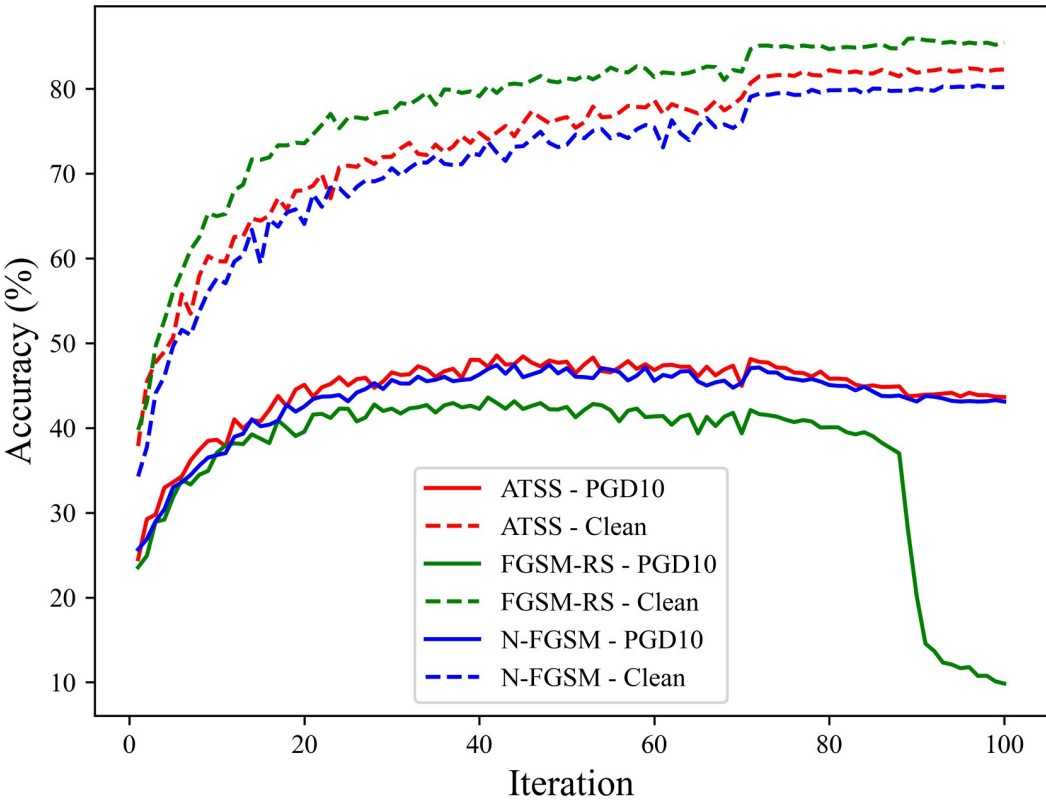

**Fig 10. Performance comparison during training among ATSS, FGSM-RS, and N-FGSM.**

**Table 4. Classification accuracy and training time under adversarial attacks for different adversarial training methods.**

| Models | Method type | Training method | Classification accuracy under different attacks(%) | | | | | | Training time (min) |
|---|---|---|---|---|---|---|---|---|---|
| | | | Clean | FGSM | PGD-10 | PGD-50 | C&W | AA | |
| ResNet18 | Multi-step | PGD-10-AT [9] | 80.22 | 52.23 | 49.12 | 48.55 | 47.81 | 45.61 | 103 |
| | | TRADES [10] | 80.71 | 52.44 | 49.67 | 48.98 | 48.13 | 45.94 | 111 |
| | Single-step | FGSM-RS [11] | 82.33 | 51.52 | 45.96 | 45.22 | 44.89 | 42.60 | **21** |
| | | N-FGSM [15] | 80.27 | 51.31 | 47.13 | 46.50 | 45.83 | 43.33 | **21** |
| | | FGSM-MEP [28] | 80.67 | 50.99 | 46.61 | 46.11 | 45.45 | 42.95 | 22 |
| | | ATAS [14] | **83.81** | 49.79 | 43.85 | 43.62 | 43.29 | 40.98 | 23 |
| | | FGSM-GA [29] | 81.46 | 50.70 | 46.67 | 46.31 | 45.71 | 43.07 | 51 |
| | | ATSS(ours) | 81.78 | **51.65** | **48.53** | **47.91** | **47.09** | **44.65** | 22 |
| VGG19 | Multi-step | PGD-10-AT [9] | 77.17 | 50.57 | 47.21 | 46.63 | 45.87 | 42.41 | 137 |
| | | TRADES [10] | 77.68 | 50.81 | 47.80 | 46.95 | 46.17 | 42.67 | 149 |
| | Single-step | FGSM-RS [11] | 79.27 | **50.03** | 44.11 | 43.21 | 42.94 | 39.23 | **31** |
| | | N-FGSM [15] | 77.21 | 49.72 | 46.21 | 45.46 | 44.58 | 40.67 | **31** |
| | | FGSM-MEP [28] | 77.61 | 49.30 | 46.07 | 45.20 | 44.22 | 40.30 | 32 |
| | | ATAS [14] | **80.75** | 48.20 | 42.44 | 41.72 | 41.24 | 37.52 | 34 |
| | | FGSM-GA [29] | 78.48 | 49.12 | 45.42 | 44.41 | 43.80 | 39.77 | 76 |
| | | ATSS(ours) | 78.92 | 49.96 | **46.60** | **45.87** | **45.02** | **41.19** | 32 |

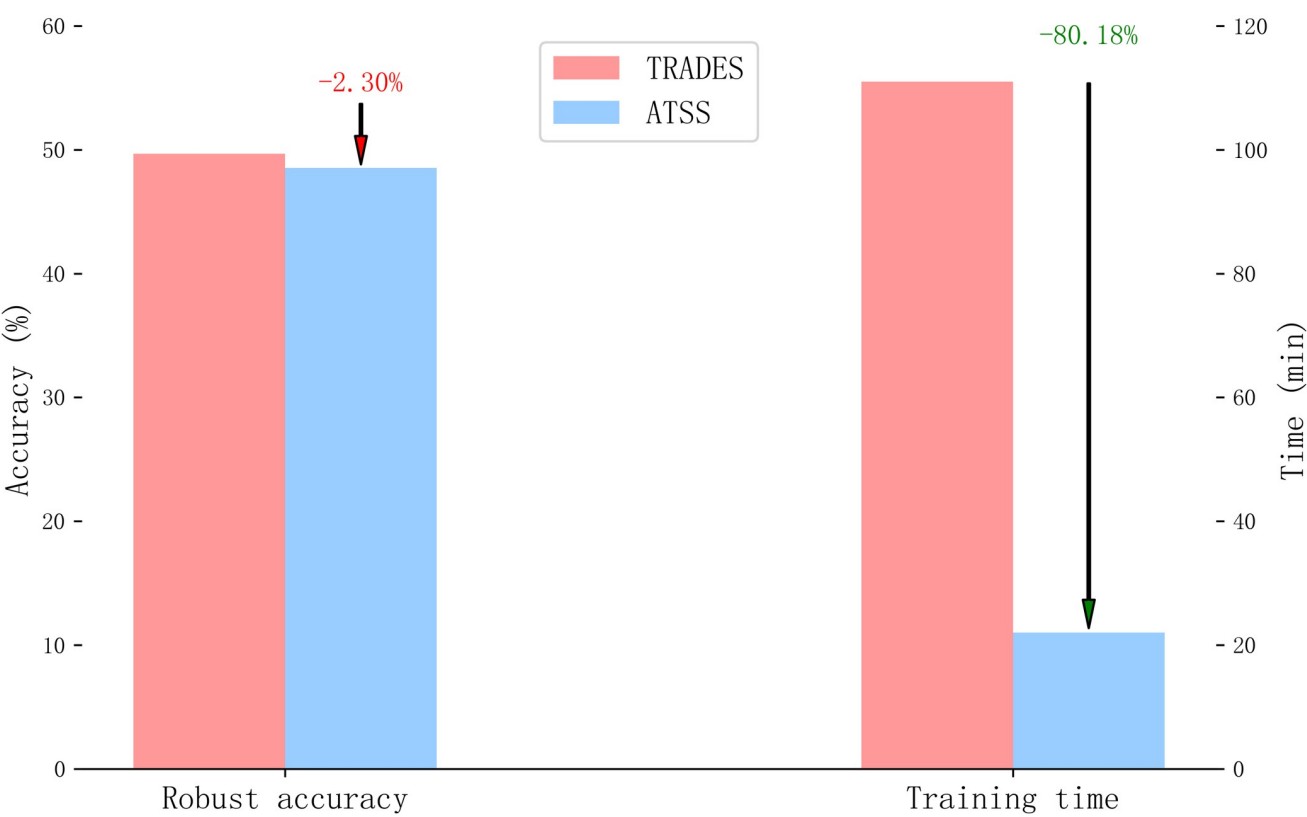

**Fig 11. Comparison of ATSS and multi-step adversarial training method in robust accuracy and training time.**

**Table 5. Classification accuracy against PGD-10 attacks on different datasets.**

| Training method | CIFAR-100 | Tiny ImageNet |
|---|---|---|
| FGSM-RS [11] | 25.12 | 19.80 |
| N-FGSM [15] | 25.71 | 20.76 |
| FGSM-MEP [28] | 25.62 | 20.52 |
| ATAS [14] | 24.78 | 19.56 |
| FGSM-GA [29] | 25.55 | 20.61 |
| ATSS(ours) | **26.11** | **21.57** |

We conduct extensive comparative experiments on various datasets, including CIFAR-10, CIFAR-100, and Tiny ImageNet, using attack methods such as FGSM, PGD, C&W, and AA to compare the proposed method with other adversarial training methods. The experimental results demonstrate that ATSS successfully addresses the catastrophic overfitting issue in fast

**Table 6. Test results under different similarity calculation strategies.**

|  | Clean acc.(%) | PGD-10 acc.(%) |
|---|---|---|
| Only $S_{cos}$ | 81.70 | 47.39 |
| Only $S_{ed}$ | 81.57 | 47.51 |
| $S_{ed} + S_{cos}$ | **81.78** | **48.53** |

**Table 7. Test results under different influence coefficient.**

| Influence coefficient | Clean acc.(%) | PGD-10 acc.(%) |
|---|---|---|
| $\beta = 0.00$ | 81.54 | 46.56 |
| $\beta = 0.02$ | 81.65 | 47.77 |
| $\beta = 0.04$ | 81.78 | **48.53** |
| $\beta = 0.06$ | 81.81 | 47.58 |
| $\beta = 0.20$ | **82.61** | 0.70 |

**Table 8. Test results under different standard step size.**

| Standard step size | Clean acc.(%) | PGD-10 acc.(%) |
|---|---|---|
| $\alpha_0 = 7/255$ | 82.17 | 45.90 |
| $\alpha_0 = 8/255$ | 82.12 | 46.54 |
| $\alpha_0 = 9/255$ | 81.94 | 47.13 |
| $\alpha_0 = 10/255$ | 81.78 | **48.53** |
| $\alpha_0 = 12/255$ | **82.35** | 0.01 |

adversarial training without significantly increasing training costs. Compared to other fast adversarial training methods, ATSS achieves better adversarial robustness while maintaining high accuracy on clean samples.

The robustness of the models trained by the proposed method still lags behind that of the current state-of-the-art multi-step adversarial training methods. In future works, more complex and sophisticated adaptive step size strategies could be explored to further enhance the model's robustness.

# Acknowledgments

The authors want to thank Dr. Zhen Xu, Dr. Yong-Bo Wu, and Dr. Huo-Ping Yi for their insights on adversarial training.

# Author Contributions

**Conceptualization:** Jie-Chao Zhao, Jin Ding.

**Funding acquisition:** Jin Ding, Ji-En Ma.

**Methodology:** Jie-Chao Zhao, Jin Ding.

**Resources:** Ping Tan.

**Software:** Jie-Chao Zhao, Yong-Zhi Sun.

**Supervision:** Jin Ding, You-Tong Fang.

**Validation:** Yong-Zhi Sun, Ping Tan.

**Visualization:** Jie-Chao Zhao, Yong-Zhi Sun, Ping Tan.

**Writing – original draft:** Jie-Chao Zhao.

**Writing – review & editing:** Jin Ding, Ji-En Ma, You-Tong Fang.

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
