## [Decision Letter · Decision Letter 0]

30 Oct 2024

PONE-D-24-43758Avoiding Catastrophic Overfitting in Fast Adversarial Training with Adaptive Similarity Step SizePLOS ONE

Dear Dr. Ding,

Thank you for submitting your manuscript to PLOS ONE. After careful consideration, we feel that it has merit but does not fully meet PLOS ONE’s publication criteria as it currently stands. Therefore, we invite you to submit a revised version of the manuscript that addresses the points raised during the review process.

The reviewers raised major comments that need to be addressed.

We look forward to receiving your revised manuscript.

Kind regards,

Alberto Marchisio

Academic Editor

PLOS ONE

3. Thank you for uploading your study's underlying data set. Unfortunately, the repository you have noted in your Data Availability statement does not qualify as an acceptable data repository according to PLOS's standards. At this time, please upload the minimal data set necessary to replicate your study's findings to a stable, public repository (such as figshare or Dryad) and provide us with the relevant URLs, DOIs, or accession numbers that may be used to access these data. For a list of recommended repositories and additional information on PLOS standards for data deposition, please see https://journals.plos.org/plosone/s/recommended-repositories.

Additional Editor Comments (if provided):

Reviewers' comments:

Reviewer's Responses to Questions

**Comments to the Author**

1. Is the manuscript technically sound, and do the data support the conclusions?

Reviewer #1: Partly

Reviewer #2: Partly

Reviewer #3: Yes

Reviewer #4: Partly

2. Has the statistical analysis been performed appropriately and rigorously? 

Reviewer #1: No

Reviewer #2: No

Reviewer #3: Yes

Reviewer #4: Yes

3. Have the authors made all data underlying the findings in their manuscript fully available?

Reviewer #1: Yes

Reviewer #2: Yes

Reviewer #3: Yes

Reviewer #4: Yes

4. Is the manuscript presented in an intelligible fashion and written in standard English?

Reviewer #1: Yes

Reviewer #2: Yes

Reviewer #3: Yes

Reviewer #4: Yes

5. Review Comments to the Author

Reviewer #1: Title: Avoiding Catastrophic Overfitting in Fast Adversarial Training with Adaptive Similarity Step Size

The article discusses strengths and weaknesses of adversarial learning when training deep neural network models. A new method implemented by the authors, called “Fast adversarial training with adaptive similarity step size (ATSS)”, is presented and tested on ResNet18 and VGG19 models by using CIFAR10, CIFAR100 and Tiny ImageNet datasets. The results confirm the validity of the approach, since ATSS avoids catastrophic overfitting, without extra computational costs

General

The article is well written and organized, the quality of English is good. I found some minor issues, which I listed below. The description of the ATSS algorithm is well structured and allows the readers to fully understand how the method works. The presentation of results on benchmark datasets could be improved in order to enhance a full comprehension: did the authors use pre-trained weights for ResNet18 and VGG19? This information is relevant. Moreover, sometimes the authors do not fully explain their outcomes (see below). Finally, the ATSS method achieves better results than other techniques with respect to accuracy, but the advantage seems limited. I understand the difficulty of achieving significantly better performances, but the authors might add some plots/statistical data supporting their claims.

Overall, the authors need to substantially modify the presentation of the experiments and their results in order to improve the quality of the article and meet the Journal requirements.

Detailed

The formatting of the paper is not justified. I suggest the authors change it.

When describing data in Tables 1 and 2, the authors do not mention the last line of both tables, that is data concerning the case “FGSM-RS(α=10/255)”. Therefore, I guess that this condition is irrelevant. Why did they insert this information?

When presenting the experimental setup, the authors state “The optimizer used in the experiments is the SGD optimizer, with a momentum of 0.9 and a weight decay coefficient of 5e-4”. Did they consider the possibility of using the Adam optimizer [1]?

[1] Kingma, D. P. & Ba, J. (2014). Adam: A method for stochastic optimization. arXiv preprint arXiv:1412.6980.

When the authors discuss the results of the experiments about prevention of catastrophic overfitting, they claim “Throughout the 100 epochs of adversarial training, the ResNet18 and VGG19 models maintain a consistent level of classification accuracy against the multi-step attack method PGD, without losing robustness due to catastrophic overfitting”. What about FGSM? They do not say anything about this technique.

Errors

Section Introduction, page 1: in the sentence “Szegedy et al. discovered that minor input perturbations, nearly imperceptible to the human eye, can cause severe classification errors in DCNNs. [4]”, move the cited reference before the dot.

Section Introduction, page 2: in the sentence “Common adversarial training approaches, such as the Projected Gradient Descent (PGD) adversarial training proposed by Madry et al. [9] and the TRADES proposed by Zhang et al. [10], employ multi-step iterative processes to generate adversarial examples, which are then used to train the model”, the acronym TRADES is not explained. I recommend the authors to add this information.

Section Introduction, page 2: in the sentence “The experimental results demonstrate that our method successfully avoids catastrophic overfitting while achieving high robustness accuracy and clean accuracy..”, please remove one of the two dots.

Section Related Work (Adversarial examples and adversarial attack methods), page 3: in the sentence “Additionally, there are optimization-based adversarial attack methods, such as C&W attack proposed by Carlini et al. [19].”, I recommend the authors to provide the meaning of the C&W acronym.

Section Related Work (Adversarial examples and adversarial attack methods), page 4: in the sentence “AutoAttack combines multiple advanced attack methods, including APGD-CE [20], APGD-DLR [20], FAB [21], and Square Attack [22], with the goal of systematically evaluating the robustness of neural network models.”, can the authors provide the meanings of the acronyms APGD-CE, APGD-DLR and FAB?

Section Related Work (Adversarial defense methods), page 5: in the sentence “Similarly, Jia et al. proposed FGSM-MEP [28], which introduces a momentum mechanism that accumulates the adversarial perturbations generated in previous iterations and uses them as the initial perturbation for the next iteration, with periodic resets after a certain number of iterations”, I recommend the authors to add the explanation of the FGSM-MEP acronym.

Section ATSS, page 8: can the authors explain the symbol f in equation 7?

Section Experiments (Datasets), page 9: in the sentence “In the experimental section, we use three publicly available benchmark datasets: CIFAR-10, CIFAR-100, and Tiny ImageNet.”, can the authors insert references to both CIFAR-10 [2], CIFAR-100 [2] and Tiny ImageNet [3] datasets? I report the references here below:

[2] Krizhevsky, A., & Hinton, G. (2009). Learning multiple layers of features from tiny images.

[3] Le, Y., & Yang, X. (2015). Tiny imagenet visual recognition challenge. CS 231N, 7(7), 3.

Section Experiments (Datasets), page 9: in the sentence “The Tiny ImageNet dataset is a smaller version of the ImageNet dataset, divided into 200 categories, containing 100,000 training images and 10,000 testing images”, can the authors insert references to the ImageNet [4] dataset? I report the references here below:

[4] Deng, J., Dong, W., Socher, R., Li, L. J., Li, K., & Fei-Fei, L. (2009). Imagenet: A large-scale hierarchical image database. In 2009 IEEE conference on computer vision and pattern recognition (pp. 248-255). IEEE.

Section Experiments (Experimental setup), page 9: in the sentence “The influence coefficient β is set to 0.04, the standard step size α0 is set to 10/255.”, the authors could add “(see section Hyperparameter Experiments)” at the end of the phrase, since they later explain how they choose these two values.

Section Experiments (Experimental setup), page 10: in the sentence “The CW adversarial attack method used is based on modifying the CW loss with PGD, with 20 iterations.”, the authors use the CW acronym, while they previously employed the C&W acronym. I recommend the authors to keep coherence in the text.

Section Experiments (Experiments on Adversarial Robustness and Training Cost), page 10: in the sentence “All methods are trained for 60 epochs, and the total training time is recorded. For robustness testing, we use five attack methods, i.e., FGSM, PGD-10, PGD-50, CW, and AA, to attack the trained models and record the classification accuracy under these

attacks.”, the authors use the CW acronym, while they previously employed the C&W acronym. Still I suggest the authors use the same acronym across the whole manuscript.

Section Experiments (Experiments on Adversarial Robustness and Training Cost), page 11: in Table 4, can the authors modify the last column from “training time” to “Training time”?

Section Conclusion, page 13: in the sentence “In future’s work, more complex and sophisticated adaptive step size strategies could be explored to further enhance the model’s robustness.”, replace “future’s work” with “future works”.

Reviewer #2: 1. There is no need to write another \\( p(\\theta) \\) in equation 3; it's more standard to place the limits of integration below the \\( E \\).

2. Is the reason the images are displayed at the end of the article due to the template?

3. In the motivation section, you attributed the magnitude of the perturbations to the main cause; have you considered the direction?

4. If you think the magnitude of the perturbations is a problem, then why not simply scale the magnitude?

Reviewer #3: The paper proposes a dynamic step to deal with catastrophic overfitting in fast adversarial training. The proposal is theoretically, with good arguments for its derivation, and practically consistent, with good support from numerical results. I believe the paper will be ready for publication after minor changes.

Specific comments:

- The algorithm 1 ATSS let the proposition very clear, but maybe it should consider the learning rate in line 9 (I believe it was used, but omitted in the algorithm) and batches instead of single images (or at least let clear what M “samples” means).

- One of the core ideas in the paper is the similarity measure, which has two components: a Euclidean and a cosine distance. In the beginning they were introduced without any good argument to be so, but it became clear that it was empirical in the results. I think this point can be better presented in the text.

- I suggest using explicit functions for Cos_{\\mean} and Cos_{\\sd}. I think it would be something like \\mean \\cos(\\eta, v) and \\std \\cos(\\eta, v).

- Do not use more than one symbol to represent one variable, parameter or adornation to not be misunderstood as multiplication of several parameters or variables (e.g. do not let “noise” in x_{noise} in italic, or use \\cos instead of Cos), specially because of uniformity since it was ok for x_{\\mean}, for instance.

- There is no need for explicit multiplication operators.

Reviewer #4: This paper introduces an enhanced adversarial training method, termed Fast Adversarial Training with Adaptive Similarity Step Size (ATSS), designed to mitigate catastrophic overfitting while improving robustness and accuracy in deep learning models. The key innovation is the use of an adaptive step size, determined by the similarity between random noise and gradient direction, to control perturbations. However, there are some improvements as follows:

1, some sections could benefit from clearer subheadings, particularly within the experimental setup, to separate discussions of model parameters, datasets, and metrics.

2, the paper would benefit from a deeper analysis of the specific limitations and cases where multi-step methods might be preferable.

3, the authors should add more flowcharts or visual explanations of the ATSS mechanism and its comparative performance, especially for non-expert readers.

4, the paper would benefit from a dedicated discussion section on potential enhancements or challenges in extending ATSS to other domains, such as speech.

6. PLOS authors have the option to publish the peer review history of their article (what does this mean?). If published, this will include your full peer review and any attached files.

Reviewer #1: No

Reviewer #2: No

Reviewer #3: No

Reviewer #4: No

---

## [Author Response · Author response to Decision Letter 0]

9 Dec 2024

Avoiding Catastrophic Overfitting in Fast Adversarial Training with Adaptive Similarity Step Size (PONE-D-24-43758)

We would like to express many thanks to reviewers and associate editor for their valuable comments, which help to enhance and improve the quality of manuscript considerably. We endeavored to address all the comments and our reflections are provided below point by point. An updated version of the paper being closed is modified based on the proposed comments. Now we summarize the responses to the reviewers’ comments as follows:

Reviewer 1:

Reviewer Comments:

Did the authors use pre-trained weights for ResNet18 and VGG19?

Authors Responses:

Thanks for your comments. ResNet18 and VGG19 are trained from the scratch. The pre-trained weights are not used. The statement has been added in the revised manuscript for clarity (see Experiments-Model Parameters Section).

Reviewer Comments:

The authors might add some plots/statistical data supporting their claims.

Authors Responses:

Thanks for your comments. Fig. 5 is added for illustrating why we need to consider both Euclidean distance similarity and cosine similarity. Fig. 8 and Fig. 9 are added for demonstrating the adversarial training process on the CIFAR-100 dataset and Tiny ImageNet dataset. Fig. 10 is added for comparing training performance among ATSS, FGSM-RS, and N-FGSM. Fig. 11 is added for comparing ATSS and TRADES in robust accuracy and training time. The added figures are marked in red.

Reviewer Comments:

The formatting of the paper is not justified. I suggest the authors change it.

Authors Responses:

Thanks for your comments. In Experiments Section, “model parameters”, “attack methods”, “baselines”, and “metrics” subheadings have been added for clarity (marked in red). 

Reviewer Comments:

When describing data in Tables 1 and 2, the authors do not mention the last line of both tables, that is data concerning the case “FGSM-RS(α=10/255)”. Therefore, I guess that this condition is irrelevant. Why did they insert this information?

Authors Responses:

Thanks for your comments. In Tables 1 and 2, we set two different step sizes, 8/255 and 10/255, for the FGSM-RS method. As we can see, under these two different step size settings, both the attack success rate and average perturbation magnitude of FGSM-RS are lower than those of the original FGSM. In the 3rd and 4th paragraph of “Motivation” subsection, the statement of the step size settings of FGSM-RS has been added for clarity (marked in red).

Reviewer Comments:

When presenting the experimental setup, the authors state “The optimizer used in the experiments is the SGD optimizer, with a momentum of 0.9 and a weight decay coefficient of 5e-4”. Did they consider the possibility of using the Adam optimizer?

Authors Responses:

Thanks for your comments. The authors considered the Adam optimizer in their experiments and tested its performance with SGD. The results, as shown in the tables below, indicate that the Adam optimizer did not perform better than SGD. Therefore, SGD was used in the experiments.

Table R1 Classification accuracy of Adam and SGD on both ResNet18 and VGG19 models on CIFAR-10 dataset

 ResNet18 VGG19

SGD 48.53% 46.60%

Adam 46.72% 42.02%

Table R2 Classification accuracy of Adam and SGD on both CIFAR-100 and Tiny ImageNet datasets on ResNet18 model

 CIFAR-100 Tiny ImageNet

SGD 26.11% 21.57%

Adam 24.73% 19.27%

Reviewer Comments:

When the authors discuss the results of the experiments about prevention of catastrophic overfitting, they claim “Throughout the 100 epochs of adversarial training, the ResNet18 and VGG19 models maintain a consistent level of classification accuracy against the multi-step attack method PGD, without losing robustness due to catastrophic overfitting”. What about FGSM? They do not say anything about this technique.

Authors Responses:

Thanks for your comments. When attacked by FGSM, both the ResNet18 and VGG19 models also can maintain a consistent level of classification accuracy. The statement has been revised in the second paragraph of “Experiments on Preventing Catastrophic Overfitting” subsection (marked in red). 

Reviewer Comments:

Section Introduction, page 1: in the sentence “Szegedy et al. discovered that minor input perturbations, nearly imperceptible to the human eye, can cause severe classification errors in DCNNs. [4]”, move the cited reference before the dot.

Authors Responses:

Thanks for your comments. The cited reference has been moved before the dot in the revised manuscript (marked in red). 

Reviewer Comments:

Section Introduction, page 2: in the sentence “Common adversarial training approaches, such as the Projected Gradient Descent (PGD) adversarial training proposed by Madry et al. [9] and the TRADES proposed by Zhang et al. [10], employ multi-step iterative processes to generate adversarial examples, which are then used to train the model”, the acronym TRADES is not explained. I recommend the authors to add this information.

Authors Responses:

Thanks for your comments. The acronym TRADES has been explained in the revised manuscript (marked in red).

Reviewer Comments:

Section Introduction, page 2: in the sentence “The experimental results demonstrate that our method successfully avoids catastrophic overfitting while achieving high robustness accuracy and clean accuracy..”, please remove one of the two dots.

Authors Responses:

Thanks for your comments. One of two dots has been removed in the revised manuscript (marked in red). 

Reviewer Comments:

Section Related Work (Adversarial examples and adversarial attack methods), page 3: in the sentence “Additionally, there are optimization-based adversarial attack methods, such as C&W attack proposed by Carlini et al. [19].”, I recommend the authors to provide the meaning of the C&W acronym.

Authors Responses:

Thanks for your comments. C&W acronym has been explained in the revised manuscript (marked in red). 

Reviewer Comments:

Section Related Work (Adversarial examples and adversarial attack methods), page 4: in the sentence “AutoAttack combines multiple advanced attack methods, including APGD-CE [20], APGD-DLR [20], FAB [21], and Square Attack [22], with the goal of systematically evaluating the robustness of neural network models.”, can the authors provide the meanings of the acronyms APGD-CE, APGD-DLR and FAB?

Authors Responses:

Thanks for your comments. The acronyms APGD-CE, APGD-DLR and FAB have been explained in the revised manuscript (marked in red). 

Reviewer Comments:

Section Related Work (Adversarial defense methods), page 5: in the sentence “Similarly, Jia et al. proposed FGSM-MEP [28], which introduces a momentum mechanism that accumulates the adversarial perturbations generated in previous iterations and uses them as the initial perturbation for the next iteration, with periodic resets after a certain number of iterations”, I recommend the authors to add the explanation of the FGSM-MEP acronym.

Authors Responses:

Thanks for your comments. The acronym FGSM-MEP has been explained in the revised manuscript (marked in red).

Reviewer Comments:

Section ATSS, page 8: can the authors explain the symbol f in equation 7?

Authors Responses:

Thanks for your comments. f represents the target model. The explanation has been added below Equation 7 (marked in red).

Reviewer Comments:

Section Experiments (Datasets), page 9: in the sentence “In the experimental section, we use three publicly available benchmark datasets: CIFAR-10, CIFAR-100, and Tiny ImageNet.”, can the authors insert references to both CIFAR-10 [2], CIFAR-100 [2] and Tiny ImageNet [3] datasets?

Section Experiments (Datasets), page 9: in the sentence “The Tiny ImageNet dataset is a smaller version of the ImageNet dataset, divided into 200 categories, containing 100,000 training images and 10,000 testing images”, can the authors insert references to the ImageNet [4] dataset?

Authors Responses:

Thanks for your comments. The authors have inserted references the reviewer provided to CIFAR-10, CIFAR-100, Tiny ImageNet, and ImageNet datasets (marked in red).

Reviewer Comments:

Section Experiments (Experimental setup), page 9: in the sentence “The influence coefficient β is set to 0.04, the standard step size α0 is set to 10/255.”, the authors could add “(see section Hyperparameter Experiments)” at the end of the phrase, since they later explain how they choose these two values.

Authors Responses:

Thanks for your comments. The phrase “(see Section Hyperparameter Experiments)” has been added following the sentence “The influence coefficient β is set to 0.04, the standard step size α0 is set to 10/255.” (marked in red).

Reviewer Comments:

Section Experiments (Experimental setup), page 10: in the sentence “The CW adversarial attack method used is based on modifying the CW loss with PGD, with 20 iterations.”, the authors use the CW acronym, while they previously employed the C&W acronym. I recommend the authors to keep coherence in the text.

Section Experiments (Experiments on Adversarial Robustness and Training Cost), page 10: in the sentence “All methods are trained for 60 epochs, and the total training time is recorded. For robustness testing, we use five attack methods, i.e., FGSM, PGD-10, PGD-50, CW, and AA, to attack the trained models and record the classification accuracy under these attacks.”, the authors use the CW acronym, while they previously employed the C&W acronym. Still I suggest the authors use the same acronym across the whole manuscript.

Authors Responses:

Thanks for your comments. All “CW” acronyms have been changed to “C&W” (marked in red).

Reviewer Comments:

Section Experiments (Experiments on Adversarial Robustness and Training Cost), page 11: in Table 4, can the authors modify the last column from “training time” to “Training time”?

Authors Responses:

Thanks for your comments. “training time” has been modified to “Training time” in Table 4 (marked in red).

Reviewer Comments:

Section Conclusion, page 13: in the sentence “In future’s work, more complex and sophisticated adaptive step size strategies could be explored to further enhance the model’s robustness.”, replace “future’s work” with “future works”.

Authors Responses:

Thanks for your comments. “future’s work” has been replaced with “future works” in Section Conclusion (marked in red).

Reviewer 2:

Reviewer Comments:

There is no need to write another \\( p(\\theta) \\) in equation 3; it's more standard to place the limits of integration below the \\( E \\).

Authors Responses:

Thanks for your comments. Equation 3 has been modified for clarity (marked in red).

Reviewer Comments:

Is the reason the images are displayed at the end of the article due to the template?

Authors Responses:

Thanks for your comments. Yes.

Reviewer Comments:

In the motivation section, you attributed the magnitude of the perturbations to the main cause; have you considered the direction? If you think the magnitude of the perturbations is a problem, then why not simply scale the magnitude?

Authors Responses:

Thanks for your comments. Adding random noise for initialization is a common-used strategy in fast adversarial training to avoid catastrophic overfitting. In FGSM-RS, adversarial examples are generated by adding noise for initialization and then applying clip operation. The adversarial examples generated by this method have a lower average perturbation magnitude compared to the original FGSM method, resulting in a lower attack success rate than the original FGSM method, as shown in Tables 1 and 2. In N-FGSM, adversarial examples are generated by adding noise for initialization but not applying clip operation. Although the adversarial examples generated by this method have a higher maximum perturbation magnitude compared to the original FGSM method, as shown in Table 3, the models trained with this method have a lower classification accuracy on clean examples, as shown in Table 4.

The reason for insufficient performance of FGSM-RS and N-FGSM is that, both methods do not consider magnitude and direction simultaneously. In the proposed ATSS, the adversarial examples are generated by computing the direction similarity and magnitude similarity between the added noise and the gradient simultaneously. Experiments show this method has better performance compared to existing fast adversarial training methods.

Reviewer 3:

Reviewer Comments:

The algorithm 1 ATSS let the proposition very clear, but maybe it should consider the learning rate in line 9 (I believe it was used, but omitted in the algorithm) and batches instead of single images (or at least let clear what M “samples” means).

Authors Responses:

Thanks for your comments. Algorithm 1 has been modified to include the learning rate and batches instead of single images (marked in red).

Reviewer Comments:

One of the core ideas in the paper is the similarity measure, which has two components: a Euclidean and a cosine distance. In the beginning they were introduced without any good argument to be so, but it became clear that it was empirical in the results. I think this point can be better presented in the text.

Authors Responses:

Thanks for your comments. Before introducing the similarity measure, the importance of combining Euclidean distance and cosine distance has been illustrated in Fig. 5 and explained. The 3rd paragraph of Section ATSS has been revised and marked in red.

Reviewer Comments:

I suggest using explicit functions for Cos_{\\mean} and Cos_{\\sd}. I think it would be something like \\mean \\cos(\\eta, v) and \\std \\cos(\\eta, v).

Authors Responses:

Thanks for your comments. Equation 10 and Equation 11 have been modified accordingly (marked in red).

Reviewer Comments:

Do not use more than one symbol to represent one variable, parameter or adornation to not be misunderstood as multiplication of several parameters or variables (e.g. do not let “noise” in x_{noise} in italic, or use \\cos instead of Cos), specially because of uniformity since it was ok for x_{\\mean}, for instance.

Authors Responses:

Thanks for your comments. Italic “noise” in Equation 6, 7, and 14 have been modified to non-italic (marked in red).

Reviewer Comments:

There is no need for explicit multiplication operators.

Authors Responses:

Thanks for your comments. Equation 9 has been modified accordingly (marked in red).

Reviewer 4:

Reviewer Comments:

some sections could benefit from clearer subheadings, particularly within the experimental setup, to separate discussions of model parameters, datasets, and metrics.

Authors Responses:

Thanks for your comments. “Experimental Setup” has been separated into “Model Parameters”, “Attack Methods”, “Baselines”, and “Metrics” (marked in red).

Reviewer Comments:

the paper would benefit from a deeper analysis of the specific limitations and cases where multi-step methods might be preferable.

Authors Responses:

Thanks for your comments. Fig. 11 has been added to illustrate the robust accuracy and training time of ATSS and multi-step method—TRADES. From the figure, it can be seen that, ATSS demonstrates comparable robust accuracy and much faster training process compared to TRADES, saving more than 80% training time. The 4th paragraph in Section “Experiments on Adversarial Robustness and Training Cost” has been marked in red.

Reviewer Comments:

the authors should add more flowcharts or visual explanations of the ATSS mechanism and its comparative performance, especially for non-expert readers.

Authors Responses:

Thanks for your comments. Fig. 5 is added for illustrating why we need to consider both Euclidean distance similarity and cosine similarity. Fig. 8 and Fig. 9 are added for demonstrating the adversarial training process on the CIFAR-100 dataset and Tiny ImageNet dataset. Fig. 10 is added for comparing training performance among ATSS, FGSM-RS, and N-FGSM. Fig. 11 is added for comparing ATSS and TRADES in robust accuracy and training time. The added figures are marked in red.

Reviewer Comments:

the paper would benefit from a dedicated discussion section on potential enhancements or challenges in extending ATSS to other domains, such as speech.

Authors Responses:

Thanks for your comments. The Discussion Section has been ad

---

## [Decision Letter · Decision Letter 1]

20 Dec 2024

Avoiding Catastrophic Overfitting in Fast Adversarial Training with Adaptive Similarity Step Size

PONE-D-24-43758R1

Dear Dr. Ding,

We’re pleased to inform you that your manuscript has been judged scientifically suitable for publication and will be formally accepted for publication once it meets all outstanding technical requirements.

Kind regards,

Alberto Marchisio

Academic Editor

PLOS ONE

Additional Editor Comments (optional):

Reviewers' comments:

Reviewer's Responses to Questions

**Comments to the Author**

1. If the authors have adequately addressed your comments raised in a previous round of review and you feel that this manuscript is now acceptable for publication, you may indicate that here to bypass the “Comments to the Author” section, enter your conflict of interest statement in the “Confidential to Editor” section, and submit your "Accept" recommendation.

Reviewer #1: All comments have been addressed

Reviewer #2: All comments have been addressed

Reviewer #3: All comments have been addressed

Reviewer #4: All comments have been addressed

2. Is the manuscript technically sound, and do the data support the conclusions?

Reviewer #1: Yes

Reviewer #2: Yes

Reviewer #3: Yes

Reviewer #4: Yes

3. Has the statistical analysis been performed appropriately and rigorously? 

Reviewer #1: N/A

Reviewer #2: No

Reviewer #3: Yes

Reviewer #4: Yes

4. Have the authors made all data underlying the findings in their manuscript fully available?

Reviewer #1: Yes

Reviewer #2: Yes

Reviewer #3: Yes

Reviewer #4: Yes

5. Is the manuscript presented in an intelligible fashion and written in standard English?

Reviewer #1: Yes

Reviewer #2: Yes

Reviewer #3: Yes

Reviewer #4: Yes

6. Review Comments to the Author

Reviewer #1: The authors have addressed all the comments raised in the reviews. The manuscript is now clearer and readers can appreciate the work. The presentation of the results has improved too. I recommend the article for publication.

Reviewer #2: (No Response)

Reviewer #3: All my concerns have been addressed. I believe the paper is now ready for publication. A minor suggestion is to define operators for mean and std, so that they do not appear in italic inside equations.

Reviewer #4: The authors have addressed my concerns well,

7. PLOS authors have the option to publish the peer review history of their article (what does this mean?). If published, this will include your full peer review and any attached files.

Reviewer #1: No

Reviewer #2: No

Reviewer #3: No

Reviewer #4: No

---

## [Editor Report · Acceptance letter]

27 Dec 2024

PONE-D-24-43758R1 

PLOS ONE

Dear Dr. Ding, 

I'm pleased to inform you that your manuscript has been deemed suitable for publication in PLOS ONE. Congratulations! Your manuscript is now being handed over to our production team.

Kind regards, 

on behalf of

Dr. Alberto Marchisio 

Academic Editor

PLOS ONE